# DECOUPLING TASK-SOLVING AND OUTPUT FORMATTING IN LLM GENERATION

## ABSTRACT

Large language models (LLMs) are increasingly adept at following instructions containing task descriptions to solve complex problems, such as mathematical reasoning and automatic evaluation (LLM-as-a-Judge). However, as prompts grow more complex, models often struggle to adhere to all instructions. This difficulty is especially common when instructive prompts intertwine reasoning directives—specifying what the model should solve—with rigid formatting requirements that dictate how the solution must be presented. The entanglement creates competing goals for the model, suggesting that more explicit separation of these two aspects could lead to improved performance. To this front, we introduce DECO-G, a decoding framework that explicitly decouples format adherence from task solving. DECO-G handles format compliance with a separate tractable probabilistic model (TPM), while prompts LLMs with only task instructions. At each decoding step, DECO-G combines next token probabilities from the LLM with the TPM calculated format compliance likelihood to form the output probability. To make this approach both practical and scalable for modern instruction-tuned LLMs, we introduce three key innovations: instruction-aware distillation, a flexible trie-building algorithm, and HMM state pruning for computational efficiency. We demonstrate the effectiveness of DECO-G across a wide range of tasks with diverse format requirements, including mathematical reasoning, LLM-as-a-judge, and event argument extraction. Overall, our approach yields 1.0% to 6.0% relative gain over regular prompting practice with guaranteed format compliance.

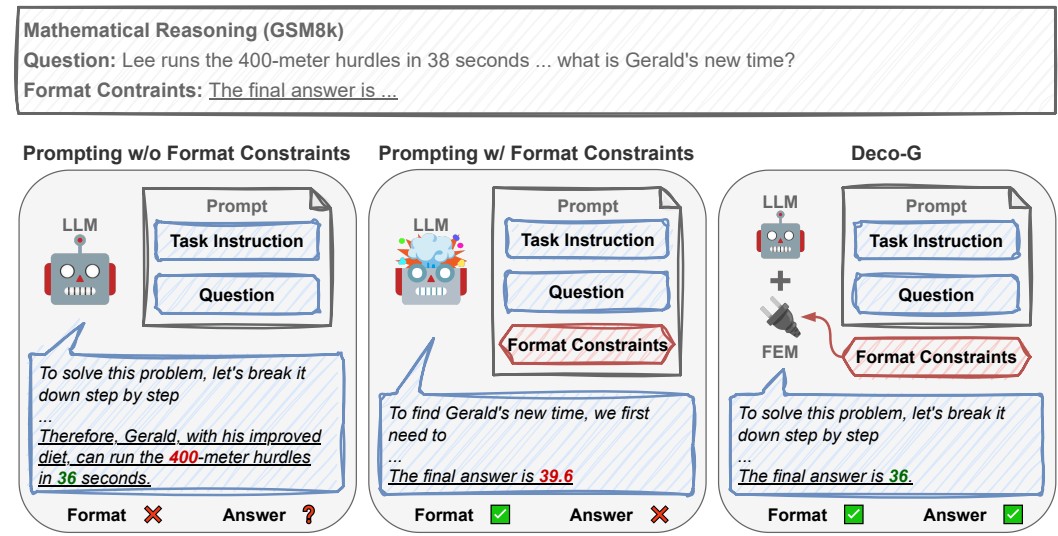

Figure 1: Example of GSM8k responses. LLM prompted without format constraints gets the correct answer, but the number is embedded in a sentence with mixed types, making it hard to capture. LLM prompted with format constraints gets the answer wrong. DECO-G prompts the model with task information and handles the format constraints by employing a Format Estimation Module (FEM). The framework generates the correct answer in the required format, making it easy to harvest.

# 1 INTRODUCTION

Instruction fine-tuning (Wei et al., 2021; Chung et al., 2024) enables large language models (LLMs) to follow user instructions and solve complex tasks. Given a task description and a desired output format, LLMs can perform tasks such as model evaluation and event extraction without additional training. Prompting strategies like Chain-of-Thought (Wei et al., 2022) and Tree-of-Thought (Yao et al., 2023) further enhance performance by encouraging structured reasoning. However, emerging evidence suggests that complex instructions—especially those with strict formatting—can negatively impact model performance (Tam et al., 2024; Long et al., 2025; He et al., 2024). For example, Long et al. (2025) reveal that LLM performance varies based on the required output format. On the MMLU (Hendrycks et al., 2020) benchmark, a model might fail to provide the correct answer when forced into one output structure, or provide the correct answer but fail a minor formatting instruction, which complicates automatic evaluation. In addition, Tam et al. (2024) point out that stricter format constraints generally lead to a greater degradation in performance on reasoning tasks. Therefore, the current paradigm of stacking task instructions and format instructions in the input prompt (as shown in Figure 1, the attachment of format instruction enclosed in the red hexagon) appears to be a limiting factor for harnessing LLM capabilities.

Attempts have been made to reduce format constraints' impact on LLM generation. For instance, Tam et al. (2024) employ a less strict format to give LLM more flexibility. Long et al. (2025) and He et al. (2024) explore formats that are more intuitive for the LLM to follow. Yet, they still pose certain constraints to the LLM, impairing its reasoning skills. Other works (Beurer-Kellner et al., 2024; guidance-ai, 2024; Willard & Louf, 2023) perform non-neural inference-time control centered toward constraint satisfaction. They guarantee format compliance by enforcing the model to decode certain tokens. This mechanism fails to consider the interplay with LLM reasoning, often resulting in incoherent output. The situation thus highlights the need for a framework that seamlessly decouples format constraints from LLM task solving to unlock the full potential of LLMs.

In this paper, we propose a decoupled generation framework DECO-G that separates output formatting from task reasoning, thereby allowing the LLM to focus on the task without the burden of format adherence. We leverage the modularity of existing controllable text generation methods (e.g. GeLaTo (Zhang et al., 2023), CtrlG (Zhang et al., 2024)) and delegate the format adherence responsibility to an auxiliary Tractable Probabilistic Model (TPM), which estimates compliance rate and reweighs token probability. While GeLaTo and CtrlG provide pathways for controllable generation with keyphrase and length constraints, they face significant challenges when applied to instruction-tuned LLMs with complex output templates. These challenges stem from a domain shift and computational bottlenecks that hinder scalability and efficiency. To make our framework practical and effective for general instructive tasks, we introduce three key techniques. Firstly, we train the HMM on LLM's instruction-response pairs to better captures task-oriented behaviors. Secondly, we employ a flexible trie-based algorithm to efficiently construct automata for complex, multi-part output templates. Thirdly, we implement HMM hidden state pruning to accelerate inference speed and ensure practical usability. To our knowledge, we are the **first** to propose the direct separation of task solving and format adherence in LLM generation to preserve its full potential.

To assess DECO-G's effectiveness in handling tasks of different natures, including reasoning and multi-phrase templates adherence, we test the framework on three different tasks: mathematical reasoning, LLM-as-a-judge evaluation, and generative event argument extraction. Experiment results show that DECO-G is able to improve overall task performance through multiple aspects: 1) improving the format satisfaction rate, 2) encouraging more natural and flexible integration of format in the output, and 3) allowing LLM to concentrate on task solving without the burden of format following. Our contributions are as follow:[1]

- We propose a framework to separate format compliance from task-solving to enhance overall performance of LLMs on various tasks through the use of a tractable probabilistic model.
- Our framework achieves high efficiency and effectiveness through technical innovations, including instruction-aware distillation, a flexible trie-building algorithm, and HMM state pruning.
- We secure improved task performance on multiple tasks compared to baseline methods, observing relative gains ranging from 1.0% to 6.0% , and provide an analysis of DECO-G's steering process with insights into its parameter setup from an entropy perspective.

---

[1]Code and model weights will be released upon paper acceptance.

## 2 PRELIMINARIES

In this section, we present our goal of task-format decoupling within a probabilistic formulation of language model generation. We further discuss how prior controllable generation methods align with this objective and provide a strong foundation for our approach.

### 2.1 GENERATION WITH ATTRIBUTE CONTROL

We frame the problem of controllable text generation using a probabilistic formulation. The auto-regressive generation of a token sequence $x_{1:n}$ given a desired attribute $\alpha$ can be expressed as:

$$P(x_{1:n}|\alpha) = \prod_t P(x_t|x_{<t}, \alpha) \tag{1}$$

The objective is to generate a sequence $x_{1:n}$ that exhibits the attribute $\alpha$. At each generation step $t$, the target distribution for producing text with the desired attribute is $P(x_t|x_{<t}, \alpha)$. Using Bayes' rule, we can rewrite this as:

$$P(x_t|x_{<t}, \alpha) = P_{\text{LM}}(x_t|x_{<t})\frac{P_{\text{LM}}(\alpha|x_t, x_{<t})}{P_{\text{LM}}(\alpha|x_{<t})} \tag{2}$$

Here, the first term, $P_{\text{LM}}(x_t|x_{<t})$, is the language model's next-token probability, which is responsible for generating fluent and coherent content. The second term, the ratio $\frac{P_{\text{LM}}(\alpha|x_t, x_{<t})}{P_{\text{LM}}(\alpha|x_{<t})}$, acts as a control signal. It quantifies how the choice of the current token $x_t$ influences the probability that the final, complete sequence will satisfy the attribute $\alpha$. However, directly calculating this ratio is intractable, as it requires marginalizing over all possible future sequences to compute the likelihoods. Thus, a key challenge in controllable generation is to find a tractable approximation for this term.

### 2.2 ESTIMATING LIKELIHOOD OF ATTRIBUTE SATISFACTION

Recent controllable generation frameworks such as GeLaTo (Zhang et al., 2023) and Ctrl-G (Zhang et al., 2024) leverage a tractable probabilistic model (TPM) to efficiently estimate the marginal probability $P(\alpha|x_t, x_{<t})$, serving as a signal to steer an LLM's generation, following

$$P(x_t|x_{<t}, \alpha) \propto P_{\text{LM}}(x_t|x_{<t})P_{\text{TPM}}(\alpha|x_t, x_{<t}) \tag{3}$$

These approaches first distill a Hidden Markov Model (HMM) as a probabilistic approximation of the LLM and then encode logical constraints to formal structure that the HMM can reason over.

**Sequence modeling with Hidden Markov Models.** A Hidden Markov Model (HMM) is the specific type of TPM used in these frameworks, chosen for its ability to model sequential data tractably. The joint probability distribution over a sequence of observed variables (tokens, $x_{1:n}$) and a corresponding sequence of hidden state variables $z_{1:n}$, is modeled as

$$P_{\text{HMM}}(x_{\leq t}, z_{\leq t}) = P(z_1)P(x_1|z_1)\prod_{t=2}^{T} P(z_t|z_{t-1})P(x_t|z_t) \tag{4}$$

Critically, the Markov property of HMMs enables efficient probabilistic inference over all possible future sequences, a task that is intractable for langauge models. In frameworks like GeLaTo and Ctrl-G, the HMM is distilled from the LLM using samples drawn unconditionally from the LLM. This process involves training the HMM via maximum likelihood on the sampled completions, equivalent to minimizing the KL-divergence between the two models' distributions $D_{\text{KL}}(P_{\text{LM}}||P_{\text{HMM}})$.

**Formalizing Constraints with Deterministic Finite Automata.** To enforce a constraint using the HMM, the constraints must be expressed in a formal language. Zhang et al. (2024) propose representing logical constraints as Deterministic Finite Automata (DFA). A DFA is an abstract state machine that recognizes patterns in sequences. Formally, a DFA is a 5-tuple $\mathcal{D} = (Q, \Sigma, \delta, q_0, F)$, where $Q$ is a finite set of states, $\Sigma$ is the alphabet (the LLM's token vocabulary), $\delta : Q \times \Sigma \to Q$ is the transition function, $q_0 \in Q$ is the initial state, and $F \subseteq Q$ is the set of accept states. A sequence is "accepted" if it drives the machine from its initial state to an accept state; otherwise, it is "rejected." This formalism is capable of representing logical constraints including the presence of keyphrases and word counts by defining the appropriate states and transitions.

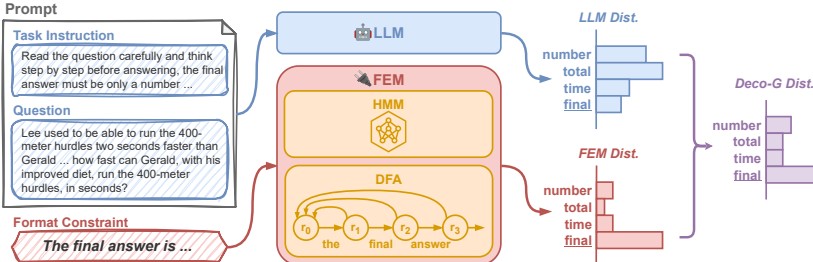

Figure 2: DECO-G decouples task and format—prompting LLM with task-only information and sending format constraints to FEM. DECO-G decodes from the posterior constructed by multiplying LLM token probabilities with FEM estimated satisfaction rate.

**Probabilistic reasoning over logical constraints.** The core idea of these prior frameworks is to use the TPM to perform a probabilistic lookahead—that is, to efficiently compute $P_{\text{TPM}}(\alpha|x_t, x_{<t})$, the probability that the full generated sequence will satisfy the constraint $\alpha$. This is accomplished by marginalizing the joint HMM-DFA state space over all possible future sequences that reach an accepting state in the DFA. According to Zhang et al. (2024), this marginalization can be calculated efficiently using a backward recurrence relation. Refer to Section B for detailed derivation.

## 2.3 FROM PRIOR WORK TO DECO-G

Prior frameworks like GeLaTo (Zhang et al., 2023) and Ctrl-G (Zhang et al., 2024) successfully use Hidden Markov Models (HMM) as tractable generative models to guide LLM generation, ensuring outputs satisfy specific logical constraints in tasks such as keyphrase generation and text editing. While GeLaTo introduces this concept, its use of Conjunctive Normal Forms (CNF) is primarily limited to keyphrase constraints. Ctrl-G generalizes this approach by specifying logical constraints through Deterministic Finite Automata (DFA), which can represent constraints on bounded-length sequences. While this foundation is promising, significant challenges arise when adapting this framework to decouple format from task reasoning for modern, instruction-tuned LLMs.

- **Domain shift:** the paradigm shifts from logical-constrained generation to separating task-instructed generation into two sub-tasks: problem solving (LLM side) and format adherence (TPM side). This incurs domain mismatch, as prior methods train HMMs on random generation without context, which is a poor proxy for an LLM conditioned on specific task instructions.
- **Complexity of format templates:** The intricate nature of real-world format templates presents a major challenge, as the overhead from constructing complex constraint automata creates a computational bottleneck.
- **Inference-time inefficiency:** The large vocabulary size of modern LLMs introduces substantial computational overhead during the inference-time guidance step , which severely impedes the framework's latency

## 3 DECO-G

In this section, we present DECO-G, a framework that realizes the decoupling of task reasoning from output formatting. As shown in Figure 2, our method separates the input prompt: the LLM receives only the task-specific information, while a dedicated Format Estimation Module (FEM) receives the format constraints. At each decoding step, the FEM estimates the likelihood of future compliance with the given format constraints $\alpha$. This likelihood is then used to reweigh the LLM's original token probabilities, steering the generation towards a format-compliant output. We now describe the key components that enable this framework.

## 3.1 INSTRUCTION-AWARE HMM DISTILLATION

An HMM can approximate a large language model's (LLM) output distribution to guide controllable generation. The fidelity of this approximation is critical—ideally, an HMM that perfectly replicates the LLM's probabilities would yield an exact posterior for format-decoupled generation, per Equa-

tion equation 2. Our key insight is that for instruction-tuned LLMs, the output distribution is fundamentally different when conditioned on a prompt versus when generating text unconditionally. Prior methods (Zhang et al., 2023; 2024), however, distill their HMMs using text sampled unconditionally from a model, an approach that fails to capture the task-oriented behavior that emerges after instruction fine-tuning. This very design renders methods like CtrlG ineffective in this context, leading to the suboptimal control patterns we demonstrate in Section C.

To bridge this gap, we carry out instruction-aware distillation: conditioning an HMM on task-oriented behavior by training it exclusively on the LLM's instruction-response pairs. Specifically, we distill knowledge from over *one million* completions generated by an LLM prompted with *one thousand* unique instructions from the Natural-Instructions-v2 (Mishra et al., 2022) dataset. Following Zhang et al. (2023), we train the HMM using the Baum-Welch algorithm (Baum et al., 1972). This process yields a robust HMM that models the LLM's conditional, instruction-following behavior, enabling more precise control over generation across a wide spectrum of tasks.

### 3.2 FLEXIBLE TRIE BUILDING FOR COMPLEX FORMAT CONSTRAINTS

To address general format constraints, we extend the DFA with an algorithm based on a flexible trie. This approach efficiently models structured templates composed of both fixed (pivots) and variable (wildcards) segments. We formally define the language of these components as follows:

- **Pivots**: a pivot $P$ is a fixed sequence of tokens $x_1 x_2 ... x_p$, representing static text in a template. The language it recognizes is a singleton set, $L_P = \{x_1 x_2 ... x_p\}$.
- **Wildcards**: a wildcard $W$ represents a *slot* to be filled by the LLM. It accepts any sequence of tokens whose length $l$ falls within a specified range $[\min, \max]$. Its language is the set of all possible strings over the alphabet $\Sigma$ within that length range:

$$L_W(\min, \max) = \bigcup_{l=\min}^{\max} \Sigma^l$$

Our flexible trie builder constructs a single DFA that recognizes a language formed by the concatenation of these components, such as $L_\alpha = L_{P_1} \cdot L_{W_1} \cdot L_{P_2} \cdot L_{W_2} \dots$. The key to its efficiency is a trie-based algorithm that shares states for all common prefixes across multiple patterns. By merging these paths into canonical representations, it constructs a compact DFA for the union of all patterns in a single pass, avoiding the state-space explosion of composing separate automata.

### 3.3 ESTIMATING FORMAT COMPLIANCE

With a distilled HMM that simulates LLM distribution and a DFA that encodes format constraints $\alpha$, we calculate the marginal probability over all sequences accepted by $\mathcal{D}(\alpha)$ as

$$P_{\text{FEM}}(\alpha | x_t, x_{<t}) = \frac{P(\mathcal{D}(\alpha) = 1, x_t, x_{<t})}{P(x_t, x_{<t})} \tag{5}$$

While the joint probability of format compliance and context sequence $P(\mathcal{D}(\alpha) = 1, x_t, x_{<t})$ is not readily available in the FEM, we follow Zhang et al. (2024)'s marginalization of HMM over DFA (Section B) to calculate this value. Finally, we use the FEM estimated compliance rate as likelihood to construct the DECO-G posterior for decoupled generation

$$P_{\text{DECO-G}}(x_t | x_{<t}, \alpha) \propto P_{\text{LM}}(x_t | x_{<t})[P_{\text{FEM}}(\alpha | x_{<t}, x_t)]^\gamma \tag{6}$$

where $\gamma$ is a hyperparameter that controls the strength of steering with default value of 1.

### 3.4 HMM HIDDEN STATE PRUNING

Although the Format Estimation Module (FEM) provides effective guidance for the generation process, its computational overhead presents a significant bottleneck during inference. The primary source of this overhead lies in the HMM's emission stage, which calculates the probability distribution over the entire vocabulary $\mathcal{V}$ from a set of $h$ hidden states. This step involves a matrix-vector multiplication with a complexity of $O(h|\mathcal{V}|)$. Given our HMM configuration with $h = 4096$ hidden states and vocabulary sizes $|\mathcal{V}|$ on the order of 128k for Llama and 152k for Qwen, this step can severely impede inference latency.

Table 1: GSM8k results.

| Method | Format (%) | Acc. (%) |
|---|---|---|
| *Llama-3.1-8B-Instruct* | | |
| NL | 96.3 | 82.3 |
| NL-S | 100 | 81.3 |
| JSON | 64.7 | 51.8 |
| JSON-S | 100 | 75.7 |
| **DECO-G** | 100 | **85.2** |
| *Qwen2.5-7B-Instruct* | | |
| NL | 98.0 | 83.6 |
| NL-S | 99.9 | 82.7 |
| JSON | 93.3 | 74.8 |
| JSON-S | 99.8 | 79.0 |
| **DECO-G$^{\gamma=2}$** | 100 | **88.6** |
| *Qwen3-8B* | | |
| NL | 97.4 | 90.5 |
| NL-S | 100 | 88.3 |
| JSON | 66.9 | 61.4 |
| JSON-S | 99.2 | 90.6 |
| **DECO-G$^{\gamma=2}$** | 100 | **91.7** |

Table 2: Generative EAE results on ACE05.

| Method | AI | AC | AI+ | AC+ |
|---|---|---|---|---|
| *Llama-3.1-8B-Instruct* | | | | |
| NL | 36.8 | 27.3 | 34.8 | 25.5 |
| NL-S | 37.1 | 27.8 | 35.1 | 26.0 |
| JSON | 35.2 | 26.2 | 33.4 | 24.6 |
| JSON-S | 33.6 | 25.2 | 31.6 | 23.6 |
| **DECO-G** | **39.4** | **28.7** | **37.0** | **26.8** |
| *Qwen2.5-7B-Instruct* | | | | |
| NL | 32.6 | 25.5 | 31.2 | 24.4 |
| NL-S | 33.2 | 24.9 | 31.1 | 23.7 |
| JSON | 31.9 | 24.1 | 30.5 | 22.9 |
| JSON-S | 34.1 | **26.1** | 32.5 | **24.7** |
| **DECO-G$^{\gamma=2}$** | **35.2** | 25.9 | **33.4** | 24.5 |
| *Qwen3-8B* | | | | |
| NL | 33.2 | **24.6** | 31.5 | **23.1** |
| NL-S | 33.3 | 24.2 | 31.5 | 22.6 |
| JSON | 31.7 | 23.4 | 3.01 | 21.8 |
| JSON-S | 32.5 | 23.0 | 30.8 | 21.5 |
| **DECO-G$^{\gamma=2}$** | **34.0** | **24.6** | **32.1** | **23.1** |

To mitigate this, we introduce **HMM hidden state pruning**, an optimization technique to reduce the computational load while preserving guidance quality. This technique is predicated on the observation that, at any given generation step, the probability mass of the hidden state distribution is concentrated within a small subset of states (see Section D). Consequently, rather than employing the full set of $h$ states for the emission probability calculation, we prune the distribution by considering only the top-k most probable states. Our empirical validation demonstrates that selecting a minimal fraction of states—specifically, the top 5% ($k = 200$) based on their probability magnitudes—is sufficient to retain over 98% of the full model's performance.

This pruning strategy drastically improves efficiency. The complexity of the emission step is reduced from $O(h|\mathcal{V}|)$ to $O(h \log h + k|\mathcal{V}|)$, where $k \ll h$. The $O(h \log h)$ term represents the cost associated with selecting the top-k states, while the dominant matrix multiplication is reduced to an $O(k|\mathcal{V}|)$ operation. This optimization achieves a considerable reduction in inference time for a negligible loss in performance, thereby enhancing the practical viability of DECO-G.

# 4 EXPERIMENT

**Experimental Setup.** We assess DECO-G's overall performance over three tasks: (1) math problem solving with reasoning, (2) LLM-as-a-judge for summary evaluation, and (3) event argument extraction as a generative task. We apply DECO-G on performant instruction models *Llama-3.1-8B-Instruct* (Grattafiori et al., 2024), *Qwen2.5-7B-Instruct* (Yang et al., 2025b), and *Qwen3-8B* (Yang et al., 2025a) to verify its effectiveness. The baselines we include for comparison are as follows,

- **NL:** prompt LLM with task instruction and natural language output constraints, free generation
- **NL-S:** prompt LLM with task instruction and natural language output constraints, structured generation enforced through *Outlines* (Willard & Louf, 2023)
- **JSON:** prompt LLM with task instruction and JSON output constraints, free generation
- **JSON-S:** prompt LLM with task instruction and JSON output constraints, structured JSON generation enforced through *Outlines*

For DECO-G, the HMM for each LLM has hidden states of size $h$=4096, output space of $|\mathcal{V}|$=128k for *Llama* and $|\mathcal{V}|$=152k for *Qwen* models, and is trained for 100 epochs on one-million LLM generated responses (sampling takes 56 GPU hours and training takes 1 GPU hour on NVIDIA A100). For the following experiments, we adopt greedy decoding to ensure fair comparison with baseline methods and evaluate zero-shot performance.

## 4.1 MATHEMATICAL REASONING

In this task, we evaluate our framework on GSM8k (Cobbe et al., 2021), a collection of grade school math problems that take two to eight steps to solve. Models are expected to carry out step-by-step reasoning and arrive at the answer. Following Tam et al. (2024), a group of task instructions is adopted to prompt the model to first reason about the math problem and then yield an integer as its answer. For JSON format output, we prompt the model to output valid JSON blob with keys *"reason"* and *"answer."* For natural language output, format instructions are used to encourage the model to generate the tamplate phrase *"The final answer is ..."* Meanwhile, this phrase is specified as a key phrase to appear in DECO-G's generation.

**Evaluation Metrics.** We measure *Format Compliance* as the rate to which the generated answer follows format requirement. In addition, we measure *Accuracy* as exact match of ground truth answer.

**Results.** As shown in Table 1, unstructured NL generation offers decent performance, with *Llama* scoring 82.3% and *Qwen* 83.6% on accuracy. However, together with unstructured JSON, free generation methods completely rely on the LLM for following the format constraint and thus suffer from low compliance rate. Structured generation, on the contrary, guarantees format compliance, but its performance is negatively impacted by the invasive intervention that sometimes cut the generation flow and alter course abruptly. DECO-G guarantees a 100% format compliance rate and achieves the best performance over all three models. In practice, we find out that *Qwen* models have more skewed token distribution. We thus raise the control factor $\lambda$ to exert stronger control on the output.

## 4.2 LLM-AS-A-JUDGE EVALUATION

We then use LLMs as judges to evaluate the quality of summaries and assess how well it aligns with human annotation. This evaluation is performed on the SummEval (Fabbri et al., 2021) dataset which consists 1600 machine-generated summaries for 100 news articles, and human annotated scores over four dimensions: Coherence, Consistency, Fluency, and Relevance. The models are asked to analyze the summary and assign a score from 1 to 5 based on the given criteria suggested by ChatGPT (OpenAI, 2025). We use the format *"The rating is ..."* for natural language output and *"rating"* as the key for harnessing JSON output.

**Evaluation Metrics.** Following Liu et al. (2023), we adopt the summary-level Spearman and Kendall-Tau correlation to gauge the performance of each method. Higher number indicates better alignment with human annotated scores.

**Results.** For this task, *Qwen* models perform well in following the output format in unstructured settings, securing over 99.7% compliance rate. This may attribute to a less intensive reasoning phase compared to mathematical reasoning. As indicated in Table 3, DECO-G demonstrates the strongest average correlation with human across models, enhancing over Consistency, Fluency, and Relevance when applied to *Qwen* models. With *Llama*, DECO-G gains over Coherence and Relevance while showing relative weakness in evaluating Consistency and Fluency. A close inspection of model generated outputs suggests that DECO-G encourages a more flexible integration of the key phrase in different places of the response: the beginning, middle, and end of response.

## 4.3 EVENT ARGUMENT EXTRACTION

The generative event argument extraction (EAE) task mainly assess a model's ability in identifying role-related arguments from source text. We evaluate on the ACE05-EN dataset (Doddington et al., 2004), in which a model is be presented with an article, a trigger word, and a set of roles to determine whether arguments associated with the roles are present in the article. This is naturally a templated task as generative model has to specify which word is extracted for which role. Regarding JSON output, we ask model to generate a JSON blob with roles as keys and extracted arguments as values. For natural language output, we specify the template *"The <role$_i$> is ..."* for every relevant roles. For DECO-G, we construct a flexible DFA that fuses the template phrases together with empty slots allowing LLM predict arguments spanning from 1 to 5 tokens.

Table 3: SummEval results, measured over Coherence, Consistency, Fluency, and Relevance.

| Method | Coherence | | Consistency | | Fluency | | Relevance | | Avg | | |
|---|---|---|---|---|---|---|---|---|---|---|---|
| | $\rho$ | $\tau$ | $\rho$ | $\tau$ | $\rho$ | $\tau$ | $\rho$ | $\tau$ | Format | $\rho$ | $\tau$ |
| *Llama-3.1-8B-Instruct* | | | | | | | | | | | |
| NL | 0.381 | 0.311 | 0.383 | 0.351 | 0.321 | 0.291 | 0.405 | 0.337 | 95.8 | 0.372 | 0.322 |
| NL-S | 0.376 | 0.308 | 0.375 | 0.343 | 0.316 | 0.287 | 0.435 | 0.364 | 100 | 0.376 | 0.325 |
| JSON | 0.449 | 0.368 | 0.446 | 0.415 | 0.326 | 0.296 | 0.424 | 0.358 | 99.8 | 0.411 | 0.359 |
| JSON-S | 0.450 | 0.369 | **0.447** | **0.416** | **0.334** | **0.302** | 0.424 | 0.358 | 100 | 0.414 | 0.361 |
| **DECO-G** | **0.458** | **0.379** | 0.439 | 0.404 | 0.331 | 0.298 | **0.441** | **0.371** | 100 | **0.418** | **0.363** |
| *Qwen2.5-7B-Instruct* | | | | | | | | | | | |
| NL | 0.407 | 0.339 | 0.442 | 0.407 | 0.291 | 0.265 | 0.399 | 0.340 | 100 | 0.385 | 0.338 |
| NL-S | 0.403 | 0.337 | 0.448 | 0.412 | 0.279 | 0.254 | 0.408 | 0.347 | 100 | 0.384 | 0.338 |
| JSON | 0.411 | 0.334 | 0.488 | 0.455 | 0.305 | 0.280 | 0.383 | 0.326 | 99.7 | 0.396 | 0.349 |
| JSON-S | **0.412** | **0.335** | 0.489 | 0.457 | 0.309 | 0.284 | 0.387 | 0.330 | 100 | 0.399 | 0.351 |
| **DECO-G**$^{\gamma=2}$ | 0.327 | 0.271 | **0.506** | **0.470** | **0.348** | **0.311** | **0.452** | **0.380** | 100 | **0.408** | **0.358** |
| *Qwen3-8B* | | | | | | | | | | | |
| NL | **0.510** | **0.416** | 0.540 | 0.504 | 0.479 | 0.441 | 0.464 | 0.392 | 100 | 0.498 | 0.439 |
| NL-S | 0.507 | 0.413 | 0.544 | 0.509 | 0.477 | 0.439 | 0.468 | 0.396 | 100 | 0.499 | 0.439 |
| JSON | 0.504 | 0.409 | 0.491 | 0.459 | 0.444 | 0.410 | 0.450 | 0.382 | 99.8 | 0.472 | 0.415 |
| JSON-S | 0.486 | 0.393 | 0.486 | 0.454 | 0.406 | 0.376 | 0.441 | 0.374 | 100 | 0.455 | 0.399 |
| **DECO-G**$^{\gamma=2}$ | 0.490 | 0.395 | **0.546** | **0.516** | **0.499** | **0.456** | **0.494** | **0.414** | 100 | **0.507** | **0.445** |

**Evaluation Metrics.** We measure performance by calculating the *f1-score* comparing the extracted tuples and the ground truth tuples for the following categories:

- Argument Id (AI): argument span and event type.
- Argument Class (AC): argument span, event type, and role type.
- Argument-attached Id (AI+): argument span, event type, and event trigger.
- Argument-attached Class (AC+): argument span, event type, event trigger, and role type.

**Results.** The *f1-scores* reported in Table 2 suggest that EAE remains a challenging task for generative models. LLMs suffer from identifying correct relations in the article and presenting valid predictions that indeed exist in the original text—without modifying entity format or referring to exterior content. Baseline methods show inconsistent trends across models, indicating LLMs' lack of robustness in event argument extraction. Employing DECO-G enhances overall extraction quality for *Llama* and *Qwen3*, while mainly improving over AI and AI+ for *Qwen2.5*. DECO-G's gain on AI and AI+ is more evident than its improvement on AC and AC+, suggesting that DECO-G can further benefit from a tighter association between roles and extracted arguments—possible through designing more natural and intuitive control phrase for DECO-G.

## 5 ANALYSIS

### 5.1 THE STEERING PROCESS

DECO-G takes advantage of HMM to estimate the future format satisfaction rate and adjust token probabilities based on the estimation. To better understand this steering process, we examine the control signals produced by the FEM and visualize the control for a span of decoding step. We track the original LLM distribution, FEM distribution, and their composed distribution, which DECO-G decodes from. Figure 3 provides an illustration of DECO-G encuraging the generation of key phrase after step by step reasoning. While the LLM tends to conclude its response with "*The total number of ... is ...*" DECO-G assigns high probabilities to the token "*final,*" steering the LLM generation to conform with format constraints.

As LLMs are trained to provide clear and concise response, they tend to avoid repeating themselves when presenting the final answer. DECO-G captures this intricacy and replaces LLM's intended conclusive phrase with the format phrase "*The final answer is*" to reduce repetition. We consider

this format integration to be more natural than forcing LLM to generate certain phrases as in regex-structured generation.

## 5.2 Token entropy and steering strength

In the previous section, we report DECO-G's results with hyperparameter $\gamma = 2$ for controlling *Qwen* models, as $\gamma = 1$ doesn't provide enough power to steer the model away from its own generation course. We hypothesize that *Qwen* models' token distributions are more skewed than *Llama*'s, making it difficult for the control signal to actually make an impact on the distribution. To verify this, we draw 100 examples from GSM8k responses and measure the average step-wise entropy of LLM token distribution. As shown in Figure 4, *Llama*'s entropy is significantly higher than those of *Qwen2.5* and *Qwen3*, suggesting that *Llama*'s token probabilities are more spread out and diverse, whereas *Qwen* models' token distributions are more peaky. This increased peakiness could be a consequence of the distribution squeezing induced by more intensive fine-tuning and preference optimization of the LLM (Ren & Sutherland, 2025). It is thus intuitive to amplify DECO-G's control strength for LLM with more skewed distribution to guarantee format compliance.

Within the same model, structured generation methods (NL-S and JSON-S) have slightly higher entropy than their unstructured counterparts (NL and JSON). This may attribute to imposed template tokens provoking more uncertainty in future token prediction. Meanwhile, DECO-G produces lowest LLM entropy, indicating that an absence of format constraint in task solving may lead to LLM providing the most confident response.

## 6 Related work

In the paper, we explore a controllable text generation (CTG) method to decouple task solving from format adherence. There are two branches in CTG that provide avenues for achieving this format-task decoupling—content-wise hard control and attribute-wise soft control.

Content-wise structured generation aims to produce outputs that conform to predefined schemes or templates. The guaranteed adherence to specified format ensures high reliability when integrating LLM with external systems. This line of methods (Willard & Louf, 2023; guidance-ai, 2024), however, exerts invasive control over the LLM generation which often produce abrupt cut-off, resulting in incomplete and incoherent responses.

Attribute-wise soft control offers a more flexible paradigm, focusing on conditioning the generation based on a desired attribute. One line of works instills attribute information into the the LLM and updates model weights, through retraining (Keskar et al., 2019; Arora et al., 2022), fine-tuning (Wei et al., 2021; Zeldes et al., 2020; Li & Liang, 2021; Lester et al., 2021), or reinforcement learning (Ouyang et al., 2022; Stiennon et al., 2020; Zeng et al., 2024; Dai et al., 2024). This method benefits from no added computational load during inference, but the expense of training the LLM for updates can be significant. The other set of works (Dathathri et al., 2019; Yang & Klein, 2021; Krause et al., 2021; Schick et al., 2021; Liu et al., 2021; Khandelwal et al., 2021; Sitdikov et al., 2022; Wen et al., 2023; Deng & Raffel, 2023) keeps the LLM as-is and instead modifies the generation probabilities at inference time, also known as *weighted decoding*. These methods typically train a lightweight auxiliary model to guide the LLM's generation at decoding time according to Bayes' rule. In light of these prior works, DECO-G takes the weighted decoding measure to compute the posterior given format constraints as an attribute.

## 7 Conclusion

In this paper, we present DECO-G, a novel decoding framework designed to decouple the responsibilities of task reasoning and format adherence. It achieves this responsibility separation by employing an auxiliary Format Estimation Module to estimate future format satisfaction and modify token probabilities, thus allowing the LLM to concentrate solely on problem-solving. Experiments on mathematical reasoning, LLM-as-a-judge evaluation, and event argument extraction demonstrate this decoupling approach leads to overall performance gain, attributing to improved format compliance, more natural format integration, and more confident response from the LLM. Limitation of this work is covered in Section A.

## ETHICAL CONSIDERATIONS

We conduct experiments on mathematical reasoning, LLM-as-a-judge evaluation, and event argument extraction. The score assigned by an LLM should not be considered an accurate reflection of quality of the summary. In addition, the LLM responses to the GSM8k questions should not be referenced for math instruction as they may include hallucination.

We acknowledge the use of AI assistants for improving the manuscript's prose, generating tables in LaTeX format, figure design, and assisting with code implementation for the analysis of HMM hidden state pruning. All generated content, particularly the data in tables, was manually verified for accuracy against our experimental results.

## REPRODUCIBILITY STATEMENT

A detailed description of our experimental setup, including the specific models used, HMM training parameters, and decoding strategy, is provided in the introductory paragraph of Section 4. To allow for replication of our experiments, the full prompts used for the mathematical reasoning (GSM8k), LLM-as-a-judge (SummEval), and event argument extraction (ACE05) tasks are detailed in Section F. Regarding computational overhead, a breakdown of the FLOPs required for the HMM forward pass, both with and without pruning, is presented in Section E. Code and model weights will be made publicly available upon paper acceptance.

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

## APPENDIX

## A  LIMITATIONS

While we show DECO-G enhances LLM task performance in various tasks, a few limitations should be taken into consideration when using DECO-G. Firstly, the HMM used to estimate format satisfaction rate is specific to an LLM, meaning that one has to distill a new HMM when switching to a different LLM. Although, in practice, we find out that an HMM can be applied to larger LLMs in the same family, it is not an accurate representation of their token distributions. Secondly, similar to other CTG methods that includes additional module for attribute modeling, DECO-G introduces additional computation overhead during decoding. As suggested by (Zhang et al., 2024), the probabilistic traversal of future generation courses using HMM has a complexity that is linear to the number of edges in DFA and quadratic to the number of hidden states in HMM. Complex format constraints, converted to larger DFA, are thus likely to increase generation runtime. Finally, finding the optimal hyperparameter $\gamma$ for LLMs with highly peaked token distributions may require empirical explorations. Such distributions require increased $\gamma$ to ensure robust format compliance, yet an excessive value may adversely affect the output quality.

## B  PROBABILISTIC REASONING OVER LOGICAL CONSTRAINTS

Zhang et al. (2023) and Zhang et al. (2024) use a TPM to perform a probabilistic lookahead—that is, to efficiently compute $P_{\text{TPM}}(\alpha|x_{\leq t})$, the probability that the full generated sequence will satisfy the constraint $\alpha$. The constraint is encoded as a DFA, and compliance is denoted by the event $\mathcal{D}(\alpha) = 1$. Then, the marginal probability over all sequences accepted by $\mathcal{D}(\alpha)$ is expressed as

$$P_{\text{TPM}}(\alpha|x_{\leq t}) = \frac{P_{\text{TPM}}(\mathcal{D}(\alpha) = 1, x_{\leq t})}{P_{\text{HMM}}(x_{\leq t})} \qquad (7)$$

where the numerator—likelihood of satisfying the constraint given a prefix $x_{\leq t}$—is found by marginalizing over the HMM hidden states $z_t$ and the DFA states $s_t$

$$P_{\text{TPM}}(\mathcal{D}(\alpha) = 1, x_{\leq t}) = \sum_{z_t} P_{\text{TPM}}(\mathcal{D}(\alpha) = 1|z_t, s_t) P_{\text{HMM}}(z_t, x_{\leq t}) \qquad (8)$$

The conditional compliance probability $P_{\text{TPM}}(\mathcal{D}(\alpha) = 1|z_t, s_t)$, as shown in Zhang et al. (2024), is calculated using a backward recurrence relation. This sums the probabilities of all valid transitions from step $t$ to $t + 1$, weighted by the HMM's transition and emission probabilities

$$P(\mathcal{D}(\alpha) = 1|z_t, s_t) = \sum_{z_{t+1}} P(z_{t+1}|z_t) \sum_{s_{t+1}} P(\mathcal{D}(\alpha) = 1|z_{t+1}, s_{t+1}) \sum_{\delta(s_t, x_{t+1}) = s_{t+1}} P(x_{t+1}|z_{t+1}) \qquad (9)$$

The computed probability then serves as the tractable approximation of the format compliance likelihood, which is used to guide the LLM's next-token generation as shown in Equation 3.

## C  SUBOPTIMAL CONTROL FROM UNCONDITIONED HMM DISTILLATION

When an instruction-tuned model is prompted with no specific user input for unconditional sampling, it often defaults to generic conversational phrases like, "*Is there something I can help you with?*" This behavior is a byproduct of its safety and helpfulness training. An HMM distilled from thousands of such non-substantive responses learns a token distribution that is unrepresentative of the model's capabilities in actual problem-solving scenarios.

Consequently, when this poorly-approximated HMM is applied to a complex reasoning task, it provides a suboptimal control signal. The HMM, having not learned the patterns of reasoned thought, cannot accurately predict the LLM's token distribution during task execution. This leads to improper guidance that can disrupt the generation process. For example, when we applied CtrlG to the GSM8k dataset, its control mechanism prematurely forced the model to generate the required format phrase ("*The final answer is ...*"), suppressing the step-by-step reasoning necessary to solve

the problem. This resulted in an accuracy of only 60.6%, a significant drop compared to the standard natural language baseline.

The table below provides an example of this failure mode on a GSM8k problem, contrasting CtrlG's flawed output with the coherent response from DECO-G, which uses an instruction-aware HMM.

Table 4: Comparison of CtrlG and DECO-G outputs on a GSM8k reasoning task.

| Method | Output |
|--------|--------|
| **CtrlG** | The final answer is 0.36 **(INCORRECT)** |
| **DECO-G** | To find the probability that both tickets are winners, we need to multiply the probabilities of each ticket winning. |

  1. The probability of the first ticket winning is 20% or 0.2.
  2. The probability of the second ticket winning is three times more likely, so it's $3 \times 0.2 = 0.6$.
  3. The probability of both tickets winning is the product of their individual probabilities: $0.2 \times 0.6 = 0.12$.
  4. To express this as a percentage, we multiply by 100: $0.12 \times 100 = 12\%$.

The final answer is 12. **(CORRECT)**

## D   HMM PRUNING AND EFFICIENCY

As established in Section 3.4, hidden state pruning is employed to mitigate the computational overhead of the HMM. This optimization is empirically justified by the highly concentrated nature of the hidden state probability distribution, as illustrated in Figure 5. Our analysis confirms that for *Llama*- and *Qwen*-distilled HMMs, the top 5% of hidden states ($k = 200$) retain over 97.8% of the total probability mass on average.

This high mass retention translates to a negligible impact on task performance. As shown in Table 6, the accuracy degradation on the GSM8k benchmark is minimal when pruning is applied: -0.2% for *Llama*, -1.2% for *Qwen2.5*, and +0.9% for *Qwen3*. This result validates that the pruned HMM provides sufficient guidance, confirming the efficacy of the optimization.

The primary benefit of this approach is a substantial improvement in computational efficiency. By reducing the computation of the HMM emission stage, pruning achieves a 13x reduction in the FLOPs required by the HMM forward function at each decoding step (from approx. 1.08 GFLOPs to 0.08 GFLOPs for *Llama*, see Section E for calculation). When compared to the LLM's own forward pass, which requires approximately 16 GFLOPs per token (Kaplan et al., 2020), the pruned FEM's computational cost constitutes only about 0.53% of the main inference workload. This optimization renders the guidance overhead practically insignificant, thereby enhancing the viability of the DECO-G framework.

## E   HMM FORWARD COMPUTATION COST

This section details the computational cost (in FLOPs) of the HMM's forward pass. The calculation uses the HMM parameters for the *Llama* model: hidden states $h$=4096, vocabulary size $|\mathcal{V}|$=128k, and $\mathrm{top}$-k states for pruning $k$=200.

**Before Pruning**   The total cost is the sum of the state transition cost ($2h^2$) and the emission cost ($2h|\mathcal{V}|$).

$$\text{Total FLOPs} = (2 \times 4096^2) + (2 \times 4096 \times 128,000)$$
$$= (3.36 \times 10^7) + (1.05 \times 10^9) \approx \textbf{1.08 GFLOPs}$$

**After Pruning** The cost is the sum of the state transition cost and the pruned emission cost ($2k|\mathcal{V}|$).

$$\text{Total FLOPs} = (2 \times 4096^2) + (2 \times 200 \times 128{,}000)$$
$$= (3.36 \times 10^7) + (5.12 \times 10^7) \approx \textbf{0.08 GFLOPs}$$

This optimization reduces the HMM's computational overhead from 1.08 GFLOPs to 0.08 GFLOPs, a $\sim$**13x reduction** per decoding step.

## F  PROMPT CONSTRUCTION

We present the set of prompts used in the experiments. For GSM8k (Table 7), we sample from a set of task instructions and a set of format instructions to construct prompts for baseline methods. For SummEval (Table 8), we include domain specific scoring criteria in the task instructions to help LLM align better with human annotations for all methods. For ACE05 (Table 9), an event description is appended to the task instructions which further explains the event of interest.

Table 5: Full ACE05 Results.

| Model | Method | AI | | | AC | | | AI+ | | | AC+ | | |
|---|---|---|---|---|---|---|---|---|---|---|---|---|---|
| | | Precision | Recall | f1 | Precision | Recall | f1 | Precision | Recall | f1 | Precision | Recall | f1 |
| Llama-3.1-8B-Instruct | NL | 33.7 | 40.5 | 36.8 | 24.2 | 31.4 | 27.3 | 30.5 | 40.5 | 34.8 | 21.7 | 31.0 | 25.5 |
| | NL-S | 33.7 | 41.3 | 37.1 | 24.4 | 32.3 | 27.8 | 30.5 | 41.3 | 35.1 | 21.8 | 32.1 | 26.0 |
| | JSON | 30.5 | 41.8 | 35.2 | 21.5 | 33.3 | 26.2 | 27.8 | 41.7 | 33.4 | 19.6 | 33.0 | 24.6 |
| | JSON-S | 28.0 | 42.0 | 33.6 | 19.8 | 34.4 | 25.2 | 25.4 | 41.6 | 31.6 | 18.0 | 34.1 | 23.6 |
| | **DECO-G** | 39.4 | 39.3 | 39.4 | 27.3 | 30.3 | 28.7 | 35.5 | 38.5 | 37.0 | 24.5 | 29.6 | 26.8 |
| Qwen2.5-7B-Instruct | NL | 29.7 | 36.2 | 32.6 | 22.6 | 29.4 | 25.5 | 27.5 | 35.7 | 31.2 | 21.0 | 29.0 | 24.4 |
| | NL-S | 28.6 | 39.5 | 33.2 | 20.7 | 31.4 | 24.9 | 26.2 | 39.0 | 31.3 | 19.1 | 31.2 | 23.7 |
| | JSON | 29.1 | 35.2 | 31.9 | 21.3 | 27.7 | 24.1 | 27.0 | 35.0 | 30.5 | 19.7 | 27.4 | 22.9 |
| | JSON-S | 32.2 | 36.2 | 34.1 | 23.9 | 28.7 | 26.1 | 29.6 | 36.1 | 32.5 | 21.9 | 28.4 | 24.7 |
| | **DECO-G** $^{\gamma=2}$ | 30.9 | 41.0 | 35.2 | 21.8 | 32.0 | 25.9 | 28.4 | 40.5 | 33.4 | 20.1 | 31.4 | 24.5 |
| Qwen3-8B | NL | 28.0 | 40.9 | 33.2 | 19.5 | 33.3 | 24.6 | 25.6 | 40.9 | 31.5 | 17.8 | 33.0 | 23.1 |
| | NL-S | 27.9 | 41.4 | 33.3 | 18.9 | 33.6 | 24.2 | 25.4 | 41.3 | 31.5 | 17.1 | 33.2 | 22.6 |
| | JSON | 27.0 | 38.4 | 31.7 | 18.6 | 31.7 | 23.4 | 24.7 | 38.4 | 30.1 | 16.7 | 31.4 | 21.8 |
| | JSON-S | 26.8 | 41.4 | 32.5 | 17.5 | 33.4 | 23.0 | 24.5 | 41.3 | 30.8 | 15.9 | 33.2 | 21.5 |
| | **DECO-G** $^{\gamma=2}$ | 27.9 | 43.6 | 34.0 | 18.9 | 35.1 | 24.6 | 25.5 | 43.4 | 32.1 | 17.3 | 34.8 | 23.1 |

## G  MORE EAE RESULTS

In Table 2, we report the *f1-scores* for each method. In Table 5, we present the full results for our event argument extraction experiment.

## H  HMM DISTILLATION AND USAGE

For *Llama* and *Qwen*, we distill their HMMs on the LLM continuation only, since the instructions from *Natural-Instructions* are human authored and should not be considered reflecting LLM distribution. We remove the special chat tokens (e.g. <|system|>, <|user|>, etc.) from the responses for HMM to capture the natural language distribution.

We tried different inputs to the HMM, including 1) regular prompt (with chat template), 2) cleaned text prompt (without chat template), and 3) no prompt (empty string). In practice, their results are almost identical. Nonetheless, in accordance with the distillation objective, we report scores yielded from using empty input to the HMM.

## I  VISUALIZATION OF DECO-G'S STEERING PROCESS

Figure 3 shows an example of DECO-G steering *Llama*'s token probability to encourage the generation of format tokens.

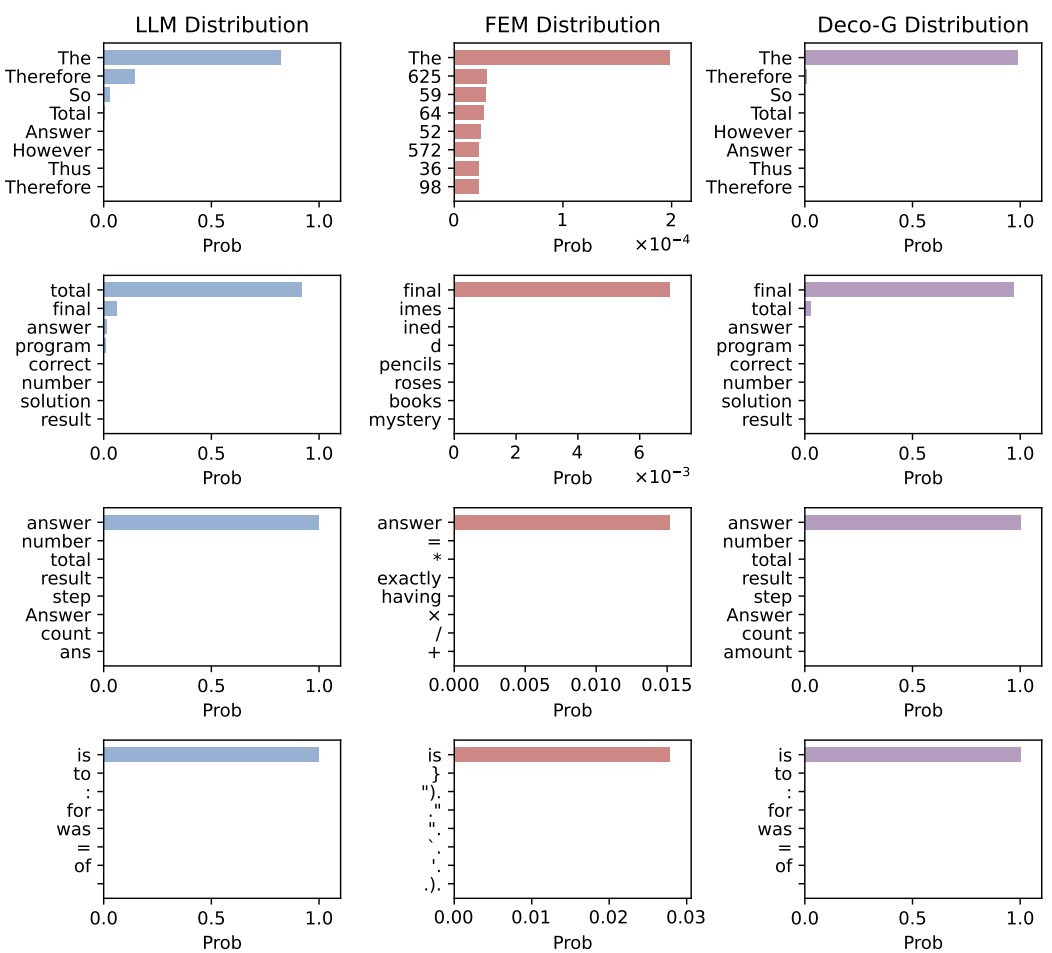

Figure 3: DECO-G steers *Llama* to generate predefined template "The final answer is ..." by boosting probabilities of template tokens.

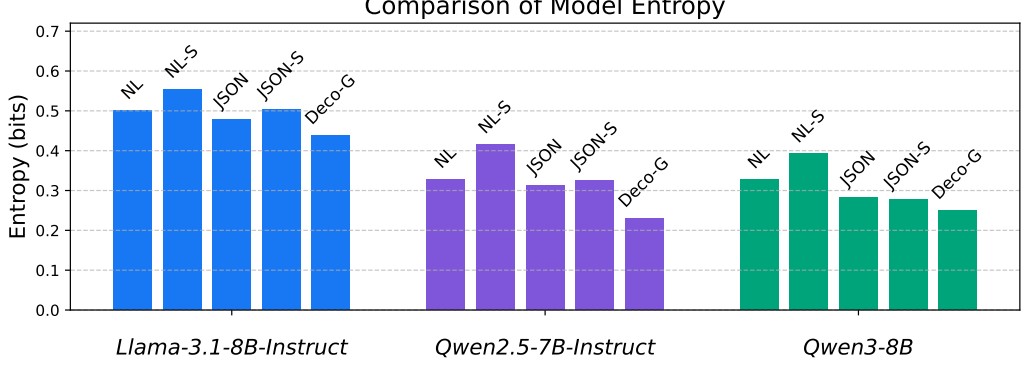

Figure 4: LLM's token-level entropy for different models and methods. *Llama* has a more flexible token distribution as compared to *Qwen*.

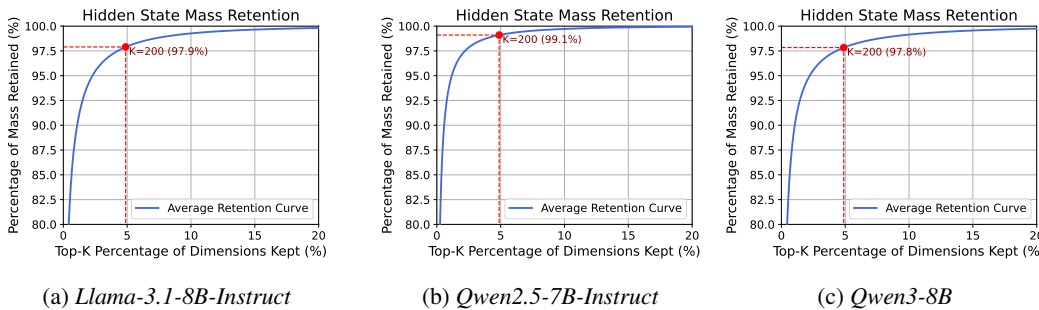

(a) *Llama-3.1-8B-Instruct*        (b) *Qwen2.5-7B-Instruct*        (c) *Qwen3-8B*

Figure 5: Average retention rate (of total mass) over top-k HMM hidden states on GSM8k dataset.

Table 6: DECO-G performance on GSM8k with and without pruning.

| Method | Acc (%) |
|---|---|
| *Llama-3.1-8B-Instruct* | |
| DECO-G w/o Pruning | 85.4 |
| DECO-G | 85.2 (Δ=-0.2) |
| *Qwen2.5-7B-Instruct* | |
| DECO-G w/o Pruning | 89.8 |
| DECO-G | 88.6 (Δ=-1.2) |
| *Qwen3-8B* | |
| DECO-G w/o Pruning | 90.8 |
| DECO-G | 91.7 (Δ=+0.9) |

Table 7: GSM8k prompt construction and an example question.

**GSM8k**

| | |
|---|---|
| **Task Instructions** | 1. Follow the instruction to complete the task:\nYou are a math tutor who helps students of all levels understand and solve mathematical problems. \nRead the last question carefully and think step by step before answering, the final answer must be only a number. |
| | 2. Follow the instruction to complete the task:\nRead the last question carefully and think step by step before answering, the final answer must be only a number. You are a math tutor who helps students of all levels understand and solve mathematical problems. |
| | 3. Follow the instruction to complete the task:\nMathematical problem-solving task:\n- Given: A mathematical question or problem\n- Required: A numerical answer only\n- Role: You are a math tutor assisting students of all levels\n- Process: Think step by step to solve the problem\nNote: Read the question carefully before beginning your analysis. |
| **NL Format Instructions** | 1. Provide your output in the following text format:\n<think step by step>. The final answer is <answer> |
| | 2. Provide your output in the following text format:\nReasoning: <reasoning first>. Answer: The final answer is ... |
| **JSON Format Instructions** | Provide your output in the following valid JSON format:\n{"reason": "<step by step reasoning>","answer": "<final answer>"} |
| **Question Example** | Janet's ducks lay 16 eggs per day. She eats three for breakfast every morning and bakes muffins for her friends every day with four. She sells the remainder at the farmers' market daily for $2 per fresh duck egg. How much in dollars does she make every day at the farmers' market? |

Table 8: SummEval prompts.

| SummEval | |
|---|---|
| **Task Instructions (Coherence)** | You will be provided with a summary written for a news article \nYour task is to rate the summary based on its coherence. \n\nPlease ensure you read and understand these instructions carefully. Keep this document open while reviewing, and refer to it as needed. \n\nEvaluation Criteria: \nCoherence (1-5): \n- 5: The summary is well-structured and organized, presenting information in a logical and seamless flow. \n- 4: The summary is mostly coherent, with minor lapses in organization or flow. \n- 3: The summary has noticeable organizational issues or lacks a smooth flow but is somewhat understandable. \n- 2: The summary is poorly structured, with significant difficulties in following its logic or flow. \n- 1: The summary is highly disjointed and lacks any meaningful structure or coherence. \nUse these criteria to assign a coherence score between 1 and 5 based on how well the summary organizes and presents information in a clear and logical manner. |
| **Task Instructions (Consistency)** | You will be provided with a news article and a summary written for this article. \nYour task is to rate the summary based on its consistency. \n\nPlease ensure you read and understand these instructions carefully. Keep this document open while reviewing, and refer to it as needed. \n\nEvaluation Criteria: \nConsistency (1-5): \n- 5: The summary is fully factually accurate and all its statements are directly supported by the source document. \n- 4: The summary is mostly factually accurate, with only minor errors or omissions. \n- 3: The summary contains noticeable factual errors or unsupported statements but retains some alignment with the source document. \n- 2: The summary has significant factual inaccuracies or includes multiple unsupported claims. \n- 1: The summary is largely inconsistent with the source, containing numerous factual inaccuracies or fabricated details. \nUse these criteria to assign a consistency score between 1 and 5 based on how well the summary aligns factually with the source article. |
| **Task Instructions (Fluency)** | You will be provided with a summary written for a news article. \nYour task is to rate the summary based on its fluency. \n\nPlease ensure you read and understand these instructions carefully. Keep this document open while reviewing, and refer to it as needed. \n\nEvaluation Criteria: \nFluency (1-5): \n- 5: The summary is clear and easy to read, with good grammar, spelling, and sentence structure. \n- 4: The summary is generally clear and fluent, with a few minor errors that don't interfere with understanding. \n- 3: The summary has some noticeable issues that might make it a little harder to read but still understandable overall. \n- 2: The summary has more noticeable problems that might make it challenging to follow in places. \n- 1: The summary has significant errors that make it difficult to read or understand in many parts. \nUse these criteria to assign a fluency score between 1 and 5 based on the quality of grammar, word choice, and sentence structure. \nImportant: When evaluating fluency, ignore punctuation and capitalization. Focus only on how natural and easy the language feels regardless of formatting. |
| **Task Instructions (Relevance)** | You will be provided with a summary written for a news article. \nYour task is to rate the summary based on its relevance. \n\nPlease ensure you read and understand these instructions carefully. Keep this document open while reviewing, and refer to it as needed. \n\nEvaluation Criteria: \nRelevance (1-5): \n- 5: The summary includes all the important information from the source document with no redundancies or irrelevant details. \n- 4: The summary is mostly relevant, with only minor omissions or slight redundancies. \n- 3: The summary includes some important information but misses key points or has noticeable redundancies. \n- 2: The summary contains limited relevant information, with significant omissions or excessive irrelevant content. \n- 1: The summary is largely irrelevant, failing to capture the main points of the source document. \nUse these criteria to assign a relevance score between 1 and 5 based on how well the summary captures the important content from the source without including excess or redundant information. |
| **NL Format Instructions** | Provide your output in the following text format: <analyze the summary>. The rating is <a number between 1 and 5> |
| **JSON Format Instructions** | Provide your output in the following valid JSON format:\n{"analysis": "<analyze the summary>","rating": <a number between 1 and 5>} |

Table 9: ACE05 prompt construction and an example question.

| ACE05 | |
| --- | --- |
| **Task Instructions** | You are an argument extractor designed to check for the presence of arguments regarding specific roles for an event in a sentence. \nTask Description: Identify all arguments related to the role Attacker, Target, Instrument, Place, Agent in the sentence. These arguments should have the semantic role corresponding to the given event trigger by the word span between [t] and [/t]. |
| | The event of interest is Conflict:Attack. The event is related to conflict and some violent physical act. Roles of interest: Attacker, Target, Instrument, Place, Agent |
| **NL Format Instructions** | Provide your output in the following text format:\nThe <role_1> is: <extracted argument>\nThe <role_2> is: <extracted argument>\n...\nThe <role_n> is: <extracted argument> |
| **JSON Format Instructions** | Provide your output in the following valid JSON format:\n{"<role>": "<extracted argument>" for role in roles of interest} |
| **Question Example** | Text: Efforts were to continue at the United Nations Friday to find a breakthrough in the diplomatic stalemate on Iraq , with Washington warning it could bypass the Security Council and go to [t] war [/t] alone . |

