# OpenReview forum: "Decoupling Task-Solving and Output Formatting in LLM Generation"
_ICLR.cc/2026/Conference — ICLR 2026 Conference Withdrawn Submission_

### Official Review · Reviewer_26FS · 2025-10-21

**Soundness:** 3
**Presentation:** 3
**Contribution:** 2
**Rating:** 4
**Confidence:** 4

**Summary:**

This paper investigates the problem of performance degradation in large language models (LLMs) when they are prompted to both perform a task (e.g., mathematical reasoning) and format their output simultaneously. To address this, the authors propose Deco-G, a framework based on a Hidden Markov Model (HMM) that estimates a model’s adherence to the required output format and adjusts its output distribution accordingly based on this feedback. The problem investigated is generally interesting. However, several concerns still remain that are summarized in the weaknessess section.

**Strengths:**

- The problem is interesting to discuss as model format consistency is a very important aspect when dealing with LLMs.
- The problem is well presented, and the paper is generally easy to follow.

**Weaknesses:**

- **Significance of the issue:** First, the paper does not clearly establish how significant the issue of formatting-induced degradation is in practice. For example, as seen in the evaluation by the authors, many models in the 7B range achieve near-perfect formatting accuracy (e.g., 99.7% compliance in LLM as a judge task by the Qwen model with formatting typically very close to 100 as seen in Table I). This suggests that newer and larger models may have largely overcome this formatting limitation. A consistent benchmarking of perhaps smaller and bigger models would make the picture clearer on that front (especially recent models).

- **Evaluation:** The evaluation by the authors focus on single prompt answer and following by the model. One intutiive way to deal with such formatting/accuracy problem is to allow the model first to provide an unrestricted (e.g., no formatting prompt) answer to the question. Then, through a second prompt, use the provided answer and the history to format the answer in the desired way. Therefore, such multi-steps prompting should also be evaluated against Deco-G given that the latter will require separate training samples and training to get good graps of the relationship between the formatting the distribution of the tokens.

- **Reasoning models:** It will be good to also evaluate reasoning models on the considered tasks to see if the reasoning traces by the models mitigates the behavior of accuracy/formatting issue that standard instruction-tuned LLMs such as Llama go through.

Minor typo: equation 2 is repeated twice at the top of page 5.

Code: It would be beneficial for the paper also to provide the code implementation

**Questions:**

Please see the Weaknesses section.

---

> ### Author Response · Authors · 2025-11-21
>
> Thank you for reviewing our paper.
>
> **1. Significance of the issue**
>
> Newer 8B-scale models have indeed improved in format compliance, achieving very high success rates. However, the performance degradation problem persists, as showcased by our experiments on Qwen3-8B. We argue that enforcing format compliance compromises model performance regardless of the model’s ability to follow instructions. Furthermore, efficient format control remains critical for smaller models, where capabilities are strictly limited by parameter count.
>
>
> **2. Evaluation**
>
> Our experiments focus on single-turn instructed task solving. While multi-step prompting is an intuitive method to separate task solving from answer formatting, it requires an additional inference pass, incurring significant computational overhead. As shown in Appendix D, DECO-G incurs only 0.53% additional computation (FLOPs) compared to the original LLM decoding. We want to highlight that our HMM is task-agnostic, which means no additional training is required when applying it to different tasks.
>
> Nonetheless, we run the two-turn prompting experiment and present the performance comparison below. In the first turn, we prompt the model with only the task instruction and question, allowing it to answer without constraints. In the second turn, we ask the model to format its response.
>
> **Table 1: Mathematical Reasoning (GSM8k)**
> | Model | DECO-G (Acc %) | Two-Stage (Acc %) |
> | :--- | :--- | :--- |
> | Llama-3.1 | 85.2 | **85.6** |
> | Qwen2.5 | 88.6 | **89.8** |
> | Qwen3 | 91.7 | **93.6** |
>
> **Table 2: LLM-as-a-Judge (SummEval) - Correlation (Pearson $\rho$ / Kendall $\tau$)**
> | Model | Method | Coherence | Consistency | Fluency | Relevance | Avg |
> | :--- | :--- | :--- | :--- | :--- | :--- | :--- |
> | **Llama-3.1** | DECO-G | **0.458 / 0.379** | 0.439 / 0.404 | 0.331 / 0.298 | **0.441 / 0.371** | **0.418 / 0.363** |
> | | Two-Turn | 0.405 / 0.330 | **0.449 / 0.411** | **0.352 / 0.317** | 0.438 / 0.367 | 0.411 / 0.356 |
> | **Qwen2.5** | DECO-G | 0.327 / 0.271 | **0.470 / 0.506** | 0.311 / **0.348** | 0.380 / **0.452** | **0.408 / 0.358** |
> | | Two-Turn | **0.363 / 0.300** | 0.438 / 0.404 | **0.314** / 0.282 | **0.443** / 0.371 | 0.390 / 0.339 |
> | **Qwen3** | DECO-G | **0.490 / 0.395** | **0.546 / 0.516** | **0.499 / 0.456** | 0.494 / 0.414 | **0.507 / 0.445** |
> | | Two-Turn | 0.476 / 0.382 | 0.545 / 0.510 | 0.494 / 0.451 | **0.500 / 0.415** | 0.504 / 0.440 |
>
>
>
> **Table 3: Event Argument Extraction (ACE05) - F1 Scores (AI / AC / AI+ / AC+)**
> | Model | DECO-G | Two-Stage |
> | :--- | :--- | :--- |
> | Llama-3.1 | **39.4 / 28.7 / 37.0 / 26.8** | 36.9 / 28.4 / 34.6 / 26.5 |
> | Qwen2.5 | **35.2 / 25.9 / 33.4 / 24.5** | 32.2 / 25.0 / 30.5 / 23.9 |
> | Qwen3 | 34.0 / 24.6 / 32.1 / 23.1 | **34.2 / 25.7 / 32.8 / 24.6** |
>
> Overall, Deco-G achieves comparable or superior performance to the two-turn baseline, particularly in structured tasks like SummEval and ACE05, while requiring significantly less computation.
>
> **3. Reasoning models**
>
> Qwen3-8B possesses strong reasoning capabilities, yet as our results show, performance still degrades when format requirements are entangled with problem-solving. While we agree that reasoning over output templates may help a reasoning model better format its response, this requires generating additional tokens. To some extent, this is similar to the two-turn solution where the model uses extra tokens to address the format issue.
>
>
> **4. Typo and code**
>
> We thank the reviewer for catching the repeated index. We will fix this error and release the code in the final version.

---

> > ### Comment · Reviewer_26FS · 2025-11-28
> > **Response to Rebuttal**
> >
> > Thanks to the authors for the rebuttal. Although the new results are useful, there are still concerns:
> >
> > Two-turns: The authors mention that two-turns (one for generating the answer and another for formatting) incurs significant computational overhead compared to Deco-G. However, even if this is true at inference time, Deco-G requires additional training (e.g., trained on 1M LLM responses as mentioned in the experimental results). Therefore, simple baselines such as two-turn prompting remain important to consider, as they eliminate the need for extra training.
> >
> > Table 2/3 results: The results in Table 2 and Table 3 are surprising. The authors should provide justification for why the proposed method would outperform a two-turn approach, where the first turn focuses on generating the answer and the second focuses on formatting. Case studies and failure analyses would be important here, specifically detailing where two-turns fails and where Deco-G succeeds.
> >
> > Table 1: For Table 1, it would be beneficial to also report formatting consistency, so that both semantic accuracy and formatting quality can be evaluated together.

---

### Official Review · Reviewer_oEU4 · 2025-10-25

**Soundness:** 2
**Presentation:** 2
**Contribution:** 3
**Rating:** 4
**Confidence:** 3

**Summary:**

The paper presents Deco-G, a framework for ensuring structured output of LLM generations. It differs from existing work by using an HMM trained on LLM generations to predict a control signal during decoding that is multiplied with the LLM's token probabilities. The framework employs HMM hidden state pruning as an optimization to reduce the inference latency.

**Strengths:**

- The paper presents a solution to the observation in recent works that enforcing output formatting decreases LLM capabilities. This is a problem that has great impact as enforcing output format (especially JSON formatting) is the common and recommended usage of LLMs.
- The work proposes a trie-algorithm and HMM state pruning optimization that improves the overhead from constrained generation (though the degree to how much is unclear).

**Weaknesses:**

- The evaluation does not present inference time results. This is valuable information when comparing the tradeoff of different decoding techniques when enforcing constraints. It is also unclear how much the HMM state pruning optimization changes in the latency. While the Appendix has some FLOPS comparison, the concrete latency numbers is still valuable metric.
- The evaluation also does not compare against using Ctrl-G to enforce format constraints. This seems like an important baseline. There's some evaluation of using Ctrl-G in Appendix C but this should be a baseline across all benchmarks in the main paper evaluation. Additionally, the paper makes the claim that "[...] using text sampled unconditionally from a model [...] fails to capture the task-oriented behavior that emerges after instruction fine-tuning". A more rigorous and thorough evaluation of the HMM training methodologies should be presented to back this claim, rather than a one-off evaluation on GSM8K.
- The description of how the work improves over Ctrl-G is not clear. The description in Sec 2.3 is vague. It may be more beneficial to the paper to highlight the improvements Deco-G makes over Ctrl-G in Section 3, highlighting concretely what each aspect of Deco-G is from Ctrl-G and which aspect is novel.
- It is not clear how the trie-algorithm composes with the the DFA. An example would be helpful in making the contribution easier to digest. Examples of constraints that are possible with Deco-G (because of the trie-algorithm) and not possible with Ctrl-G would also be beneficial.

**Questions:**

- What are the inference latency of each method on the benchmarks?
- What is the significance of the trie-algorithm? What formats can Deco-G enforce that other methods cannot because of it? Is there a computation cost to using a trie on formats that can be enforced on all methods?
- How does Deco-G compare to Ctrl-G on enforcing format constraints on the evaluation benchmarks?

---

> ### Author Response · Authors · 2025-11-21
>
> Thank you for reviewing our paper. We appreciate your feedback and conducted additional experiments to address your concerns.
>
> **1. Inference Latency**
>
> We evaluated the inference throughput on the GSM8k benchmark (1319 questions) using Llama-3.1-8B-Instruct on a single NVIDIA A100 (batch size=32, new_max_tokens=500).
>
> | Method | Total Time | Throughput (samples/s) |
> | :--- | :--- | :--- |
> | (1) Original LLM | 13m 39s | 1.61 |
> | (2) Full HMM | 38m 57s | 0.56 |
> | (3) DECO-G (Pruned) | 19m 23s | 1.13 |
>
> Deco-G’s pruning of HMM hidden states significantly improves throughput, doubling the speed compared to the Full HMM generation (0.56 $\to$ 1.13 samples/s). While DECO-G introduces a 42% overhead compared to unconstrained decoding, it remains practical for real-world usage while guaranteeing format compliance. We believe this throughput can be further improved with kernel-level optimizations.
>
> **2. Comparison with CtrlG**
>
> As per the reviewer’s request, we conducted additional experiments using CtrlG as a baseline. We focused on CtrlG as it is a generalization of GeLaTo that supports broader logical constraints required by our tasks. Due to compute constraints during the rebuttal period, we perform this comparison using the Llama-3.1-8B-Instruct backbone.
>
> **Table 1: GSM8k Result**
> | Method | Accuracy (%) | Format (%) |
> | :--- | :--- | :--- |
> | NL | 82.3 | 96.3 |
> | NL-S | 81.3 | 100.0 |
> | JSON | 51.8 | 64.7 |
> | JSON-S | 75.7 | 100.0 |
> | **CtrlG** | 61.6 | 100.0 |
> | **DECO-G (Ours)** | **85.2** | **100.0** |
>
>
> **Table 2: SummEval Result**
> | Method | Coherence | Consistency | Fluency | Relevance | Avg |
> | :--- | :--- | :--- | :--- | :--- | :--- |
> | NL | 0.381 / 0.311 | 0.383 / 0.351 | 0.321 / 0.291 | 0.405 / 0.337 | 0.372 / 0.322 |
> | NL-S | 0.376 / 0.308 | 0.375 / 0.343 | 0.316 / 0.287 | 0.435 / 0.364 | 0.376 / 0.325 |
> | JSON | 0.449 / 0.368 | 0.446 / 0.415 | 0.326 / 0.296 | 0.424 / 0.358 | 0.411 / 0.359 |
> | JSON-S | 0.450 / 0.369 | **0.447** / **0.416** | **0.334** / **0.302** | 0.424 / 0.358 | 0.414 / 0.361 |
> | **CtrlG** | 0.449 / 0.368 | 0.440 / 0.403 | 0.327 / 0.296 | 0.437 / 0.369 | 0.413 / 0.359 |
> | **DECO-G (Ours)** | **0.458** / **0.379** | 0.439 / 0.404 | 0.331 / 0.298 | **0.441** / **0.371** | **0.418** / **0.363** |
>
>
> **Table 3: ACE05 Result**
> | Method | AI | AC | AI+ | AC+ |
> | :--- | :--- | :--- | :--- | :--- |
> | NL | 36.8 | 27.3 | 34.8 | 25.5 |
> | NL-S | 37.1 | 27.8 | 35.1 | 26.0 |
> | JSON | 35.2 | 26.2 | 33.4 | 24.6 |
> | JSON-S | 33.6 | 25.2 | 31.6 | 23.6 |
> | **CtrlG** | 39.1 | 27.5 | **37.0** | 26.1 |
> | **DECO-G (Ours)**| **39.4** | **28.7** | **37.0** | **26.8** |
>
> **Analysis:** We observe a significant performance gap in GSM8k. A close look into the generation reveals that CtrlG tends to force the model to generate the required format early in the sentence, which hinders the model from reasoning over the problem for sufficient number of tokens. Deco-G, in contrast, allows the reasoning process to unfold naturally before generating format constraints.
>
> **3. Improvements over CtrlG**
>
> We would like to first clarify the scope of our contributions, which are two-folds:
>
> * **Framework level:** we challenge the status quo of stacking task instruction and format constraints into a single prompt for generation. We propose a shift in paradigm to decouple these competing objectives, allowing the LLM to focus purely on reasoning.
> * **Technical side:** we support the shift in paradigm by proposing three innovations: instruction-aware distillation, flexible trie-building, and HMM pruning.
>
> We hope the reviewer recognizes the significance of our framework-level contribution, which complements the technical improvements proposed in the paper.
>
> Deco-G improves over CtrlG in three dimensions.
>
> 1. **Instruction-Aware Distillation:** Unlike CtrlG (unconditional), we distill on instruction-response pairs. This captures task-oriented behavior and prevents the reasoning degradation seen in GSM8k.
> 2. **Flexible Trie-Building:** CtrlG relies on standard DFA which struggle with open-ended generation slots. We introduce a Trie-based construction that efficiently handles templates with variable-length "wildcards".
> 3. **HMM Pruning:** We introduce top-$k$ state pruning to reduce the computational bottleneck, improving throughput by ~2x over the full HMM approach used in CtrlG.

---

> > ### Author Response · Authors · 2025-11-21
> >
> > **4. Flexible Trie Algorithm**
> >
> > Our flexible Trie algorithm constructs compact DFA for interleaved “pivot” and variable length “wildcard” tokens. Standard DFA construction, as in CtrlG, often struggles with the bounded variable-length constraints (e.g., generate 5-10 arbitrary tokens before getting to the next set of pivots), leading to state explosion or additional manual design for optimization. Our algorithm solves this by treating “wildcards” as specialized counting states that govern transition logic to the next pivot.
> >
> > Formats like “Role A is [1-5 tokens] Role B is [1-5 tokens] Role C is [1-5 tokens] …” require sequential control while allowing flexibility within the argument span. Simple combination of $N$ separate DFAs result in $5^N$ distinct paths, whereas in our Trie-building we maintain a linear state space ($O(N \times K)$) by merging divergent wildcard paths back into a single state at each pivot.
> >
> > **Complexity**
> >
> > **Standard Construction (early expansion):** $O(K^N)$ states.
> > * Every length variation creates a new timeline.
> > * State space explodes exponentially with the number of slots ($N$).
> >
> > **Deco-G (state convergence):** $O(N \times K)$ states.
> > * There are $K$ states for the wildcard (to count the length).
> > * There are $L_{pivot}$ states for the pivot.
> > * Because they merge back to a single point after every slot, the complexity is additive
> >
> >
> >
> > **5. Computation Cost**
> >
> > DFA incurs minimal inference cost. Once the Trie is built into a DFA, it becomes an O(1) lookup function. Computation needed for DFA construction is effectively $O(L)$ where $L$ is the template length. Therefore, applying this Trie-building algorithm incurs negligible cost in general.

---

> > > ### Comment · Reviewer_oEU4 · 2025-11-25
> > >
> > > Inference Latency, Ctrl-G: These concerns have been addressed, thank you.
> > >
> > > For the rest, I have some follow up concerns and questions.
> > >
> > > This is my understanding of the claims of this work, please correct me if I have misunderstood:
> > > 1. instruction-aware distillation improves response accuracy/quality over Ctrl-G's unconditional distillation
> > >     - The additional evaluation results with Ctrl-G validates this.
> > > 2. flexible trie-building enables efficient DFA construction (improves memory/latency) over Ctrl-G's standard DFA construction
> > >     - The theoretical analysis of the complexity of the flexible trie algorithm partially supports this. Empirical evaluation of this would better support this claim, as it could be that in most cases the output format constraints is not in the worst case regime.
> > > 3. HMM pruning improves latency over Ctrl-G's full HMM approach
> > >     - The additional evaluation results on inference latency addresses this.
> > >
> > > Then, is my understanding correct that Ctrl-G and Deco-G can (in theory) support the same output format, but Deco-G is more efficient in the cases where there are fully unconstrained tokens amidst fully constrained tokens? If that is the case, how does Ctrl-G and Deco-G compare in terms of latency/inference throughput on the benchmarked tasks (point 2)? If I am mistaken, could you clarify on what metrics Ctrl-G "struggle with open-ended generation slots"?
> > >
> > > Additionally, an example walkthrough with the prompts, output format constraints given to the FEM, example outputs, and failure of Ctrl-G and a brief explanation on why (and how Deco-G addresses the Ctrl-G's weakness on this front) should be lifted and included in the main paper to make the proposed methodology clearer.

---

### Official Review · Reviewer_mTG8 · 2025-10-31

**Soundness:** 2
**Presentation:** 3
**Contribution:** 2
**Rating:** 4
**Confidence:** 3

**Summary:**

This paper proposes DECO-G, a decoding framework that integrates learned structural format models with LLMs to improve outputs that must follow explicit schemas (e.g., JSON). The core idea is to train instruction-conditioned Hidden Markov Models (HMMs) and flexible tries/DFA-based validators that capture structural 'regularities' in model outputs. While HMMs are used in prior work, this method trains them on instruction-resonse pairs. During inference, similar to existing methods,  DECO-G combines these learned structural priors with token-level probabilities from the LLM to guide decoding — preferring sequences that are both semantically likely and structurally valid. Experiments have been presented on GSM8K for mathematical reasoning, ACE05 for event extraction and the use in an LLM as a judge for scoring summaries of news articles demonstrating better performance than using Outlines.

**Strengths:**

- Clear motivation: forcing a model to handle both reasoning and strict format at once can degrade reasoning.
 - Instructions conditioned HMM training seems like a useful idea for the task

**Weaknesses:**

- It is unclear how well custom or non-JSON formats can be learned. The mechanism for discovering pivot and wildcard boundaries is under-described, making it difficult to assess how the model generalizes beyond standard JSON structures.

- Evaluation focuses exclusively on JSON outputs; no experiments demonstrate performance on alternative structured formats such as XML, Markdown, or list-style instructions.

- The use of the SNI dataset suggests that the diversity of instruction types and output schemas may be somewhat limited

- Key implementation details are missing, which limits reproducibility and clarity.

**Questions:**

1. It is not clear how the HMM is explicitly encouraged to model structure rather than surface token sequences. If the model is learning from JSON-like data with arbitrary keys and values, some preprocessing—such as parsing with a JSON parser—would presumably be required to expose structural boundaries (e.g., braces, delimiters, key–value pairs). In that case, the induced HMM and DFA might simply encode the constructs of JSON rather than learning structure in a generalizable way. How would the approach extend to formats that are not easily tokenized, such as lists, Markdown tables, XML, or other instruction-defined layouts? There are also some format following benchmarks; a clarification of why they were not considered would be helpful as well.

2. Conversely, if no parser is used, it is unclear how the system segments raw text into pivots and wildcards without leaking task-specific information into the HMM. In that setting, wouldn’t supervised fine-tuning (SFT) on the same data serve as a natural and potentially stronger baseline?

Perhaps I am misunderstanding a few implementation details—I'd be happy to see the authors clarify these points and revise my score accordingly.

---

> ### Author Response · Authors · 2025-11-21
>
> We thank the reviewer for recognizing the motivation of our work. However, there appears to be a misunderstanding regarding the distinct roles of the HMM and the DFA. We clarify these roles below, which we believe would resolve the questions and concerns regarding the generalization of our framework.
>
> **1. Mechanism Clarification**
>
> We want to clarify that the **HMM does not learn or encode the output format**. HMM’s purpose is to model LLM’s general token distribution. It is a lightweight proxy that allows efficient lookahead, through which we can estimate the probability that the LLM generates certain completions regardless of the format. It is trained on model responses to the Natural Instructions to capture general instruction-following dynamics, not specific schemas like JSON or XML. Whereas the DFA is task-specific and explicitly encodes format constraints. It accepts only sequences that adhere to user-defined rules.
>
> **2. Clarification on Output Formats**
>
> We would like to clarify that Deco-G does not generate JSON in our experiments. It generates outputs that satisfy certain task-specific text constraints. For example, for GSM8k, it is asked to generate response with *“The final answer is …”*. We compare the performance under this setting against different prompting methods and structured generation methods.
>
> The reason we prioritize natural language templates is that rigid formats like JSON and XML degrade reasoning performance. As shown in Tam et al. [1], forcing models to output strict formats (XML/JSON/YAML) often hurts accuracy compared to natural language. Since Deco-G guarantees adherence to templates, we choose natural language templates to preserve the model's reasoning capabilities while ensuring the answer can be easily parsed.
>
> **3. Encoding Format Constraints**
>
> The “pivots” and “wildcards” mentioned in the paper are user-defined inputs based on the specific formatting needs (e.g., user input: “The rating is [WILDCARD]”). The system does not need to "segment raw text" or "discover" these boundaries. They are part of the constraint definition determined by the user.
>
> **4. SFT vs. Decoding-Time Control**
>
> SFT and our decoding time method serve different purposes. SFT adapts models to a specific task or domain, while Deco-G keeps the LLM frozen and reweights token probability in decoding time. For the goal of decoupling format constraints from reasoning, we deem decoding-time control to be better because it allows the model to handle different format constraints without retraining.
>
>
> [1] *Let Me Speak Freely? A Study on the Impact of Format Restrictions on Performance of Large Language Models*, Tam et al., 2024

---

> > ### Comment · Reviewer_mTG8 · 2025-11-22
> >
> > ***The reason we prioritize natural language templates is that rigid formats like JSON and XML degrade reasoning performance. As shown in Tam et al. [1], forcing models to output strict formats (XML/JSON/YAML) often hurts accuracy compared to natural language. Since Deco-G guarantees adherence to templates, we choose natural language templates to preserve the model's reasoning capabilities while ensuring the answer can be easily parsed.***
> >
> > But isn't the whole point of decoupling towards mitigating this problem?

---

> > > ### Author Response · Authors · 2025-11-22
> > > **Respond to Reviewer mTG8's followup**
> > >
> > > Thank you for the followup.
> > > Yes, the idea of decoupling format from reasoning is to let language model focus on reasoning, while introduce other mechanisms (in our case an HMM) to enforce format. However, there is no shown benefit of **JSON** or **XML** output format being better than natural language templates. If everything equals, forcing formats as natural language templates is easy to implement and guaranteed to parse out the final answer. In other words, we do not see benefits forcing **JSON** or **XML** output format.

---

### Official Review · Reviewer_jv6h · 2025-11-01

**Soundness:** 3
**Presentation:** 3
**Contribution:** 2
**Rating:** 4
**Confidence:** 3

**Summary:**

This paper introduces a decoding framework called DECO-G. The aim is to solve the problem that LLMs performed less well when asked to solve a certain problem (e.g., a math problem) and at the same time have format constraints. The latter is important for extracting the answer easily pragmatically. The key idea is to decouple these tasks. The framework uses a so-called traceable probabilistic model for handling formatting, whereas the input to the LLM only focuses on the task instructions. The work is based on controllable text generation (GeLaTO and CtrlG), and focuses on the problem of combining instruction-tuned LLMs with output templates. Contributions of the paper include (i) the idea of instruction-aware distillation, (ii) extending the decoding format constraints from DFA to trie, and (iii) the technique of HMM hidden state pruning.

The approach is tested on three different kinds of tasks (math reasoning, LLM-as-a-judge, and event argument extraction).

**Strengths:**

- The paper describes an interesting problem of getting good accuracy while having output constraints.

- The paper is quite easy to read and well written

- The evaluation setup with the three areas of math reasoning, LLM-as-a-judge, and event argument extraction is reasonable

**Weaknesses:**

- It is not clear that the evaluation is comparing with the right baseline. The method compares free generation (named NL) with -S extensions using Outlines by Willard & Lauf. However, the method of outlines has not been discussed in the paper before, and the closely related methods (GeLaTo and CtrlG) have not been used as baselines.

- The technical part describing the new ideas and main contributions is only briefly described in section 3.

- Only JSON is evaluated as an output constraint. Hence, it is hard to know how generalizable the output constraint restrictions are.

- The related work section does not seem to capture new related developments, e.g., "Syntactic and Semantic Control of Large Language Models via Sequential Monte Carlo" from ICLR 2025, "Fast Controlled Generation from Language Models with Adaptive Weighted Rejection Sampling" from COLM 2025.

**Questions:**

- Why is the work not compared to GeLaTo and CtrlG in the experimental evaluation?

---

> ### Author Response · Authors · 2025-11-21
>
> We thank the reviewer for the comprehensive review. We would like to take this opportunity to clarify the scope of our contributions, which are two-folds:
>
> * **Framework level:** we challenge the status quo of stacking task instruction and format constraints into a single prompt for generation. We propose a shift in paradigm to decouple these competing objectives, allowing the LLM to focus purely on reasoning.
> * **Technical side:** we support the shift in paradigm by proposing three innovations: instruction-aware distillation, flexible trie-building, and HMM pruning.
>
> We hope the reviewer recognizes the significance of our framework-level contribution, which complements the technical improvements proposed in the paper.
>
>
> **1. Comparison with Baselines (CtrlG)**
>
> Given our primary focus on the framework-level shift, we prioritize comparing against methods that took the prompting or structured route, which is, we compared over NL, NL-S, JSON, and JSON-S. As per the reviewer’s request, we conducted additional experiments using CtrlG as a baseline. We focused on CtrlG as it is a generalization of GeLaTo that supports broader logical constraints required by our tasks. Due to compute constraints during the rebuttal period, we perform this comparison using the Llama-3.1-8B-Instruct backbone.
>
>
> **Table 1: GSM8k Result**
> | Method | Accuracy (%) | Format (%) |
> | :--- | :--- | :--- |
> | NL | 82.3 | 96.3 |
> | NL-S | 81.3 | 100.0 |
> | JSON | 51.8 | 64.7 |
> | JSON-S | 75.7 | 100.0 |
> | **CtrlG** | 61.6 | 100.0 |
> | **DECO-G (Ours)** | **85.2** | **100.0** |
>
>
> **Table 2: SummEval Result**
> | Method | Coherence | Consistency | Fluency | Relevance | Avg |
> | :--- | :--- | :--- | :--- | :--- | :--- |
> | NL | 0.381 / 0.311 | 0.383 / 0.351 | 0.321 / 0.291 | 0.405 / 0.337 | 0.372 / 0.322 |
> | NL-S | 0.376 / 0.308 | 0.375 / 0.343 | 0.316 / 0.287 | 0.435 / 0.364 | 0.376 / 0.325 |
> | JSON | 0.449 / 0.368 | 0.446 / 0.415 | 0.326 / 0.296 | 0.424 / 0.358 | 0.411 / 0.359 |
> | JSON-S | 0.450 / 0.369 | **0.447** / **0.416** | **0.334** / **0.302** | 0.424 / 0.358 | 0.414 / 0.361 |
> | **CtrlG** | 0.449 / 0.368 | 0.440 / 0.403 | 0.327 / 0.296 | 0.437 / 0.369 | 0.413 / 0.359 |
> | **DECO-G (Ours)** | **0.458** / **0.379** | 0.439 / 0.404 | 0.331 / 0.298 | **0.441** / **0.371** | **0.418** / **0.363** |
>
>
> **Table 3: ACE05 Result**
> | Method | AI | AC | AI+ | AC+ |
> | :--- | :--- | :--- | :--- | :--- |
> | NL | 36.8 | 27.3 | 34.8 | 25.5 |
> | NL-S | 37.1 | 27.8 | 35.1 | 26.0 |
> | JSON | 35.2 | 26.2 | 33.4 | 24.6 |
> | JSON-S | 33.6 | 25.2 | 31.6 | 23.6 |
> | **CtrlG** | 39.1 | 27.5 | **37.0** | 26.1 |
> | **DECO-G (Ours)**| **39.4** | **28.7** | **37.0** | **26.8** |
>
> **Analysis:** We observe a significant performance gap in GSM8k. A close look into the generation reveals that CtrlG tends to force the model to generate the required format early in the sentence, which hinders the model from reasoning over the problem for sufficient number of tokens. Deco-G, in contrast, allows the reasoning process to unfold naturally before generating format constraints.
>
>
> **2. Clarifications on Baselines and Constraints**
>
> Outlines is a structured generation library that guarantees required format by compiling the constraints into a finite state machine that modifies token probabilities to mask out tokens that would violate the schema. We will add this introduction to the baseline section and clarify that -S tag indicates the use of Outline.
>
> We compare against natural language (NL) and JSON constraints because they are the most prevalent output formats in real-world applications. Results in Tam et al.[1] show that, in general, NL and JSON constraints lead to better performance as compared to other formats like XML or YAML. We thus select two constraint types that LLM performs the best as baselines.
>
>
> **3. Technical Details**
>
> For the technical part, we include (1) instruction-aware distillation, (2) flexible trie-building, and (3) HMM pruning in Section 3. More technical detail and evidence supporting our design are placed in Appendix B through D due to page limit. In the final version, we will move distillation details to the main paper for a more comprehensive presentation.
>
>
> **4. Related Work**
>
> We thank the reviewer for pointing out relevant papers on adaptive rejection sampling and sequential Monte Carlo approximation. These methods further enrich controlled generation of LLMs. We will cite and discuss both in our related work section.
>
>
> **Reference**
>
> [1] *Let Me Speak Freely? A Study on the Impact of Format Restrictions on Performance of Large Language Models*, Tam et al., 2024

---

### Note · Authors · 2025-12-29

I have read and agree with the venue's withdrawal policy on behalf of myself and my co-authors.